# What's the harm? Results of an active surveillance adverse event reporting system for chiropractors and physiotherapists

Katherine A. Pohlman[1], Martha Funabashi[2,3], Maeve O'Beirne[4], J. David Cassidy[5‡], Michael D. Hill[6‡], Eric L. Hurwitz[7‡], Gregory Kawchuk[8‡], Silvano Mior[2,9‡], Quazi Ibrahim[10‡], Haymo Thiel[11‡], Michael Westaway[12‡], Jerome Yager[13‡], Sunita Vohra[14]*

1 Research Center, Parker University, Dallas, Texas, United States of America, 2 Division of Research and Innovation, Canadian Memorial Chiropractic College, Toronto, Ontario, Canada, 3 Department of Chiropractic, Universite du Quebec a Trois-Rivieres, Trois-Rivieres, Montreal, Canada, 4 Family Medicine, University of Calgary, Calgary, Alberta, Canada, 5 Division of Epidemiology, Dalla Lana School of Public Health, University of Toronto, Toronto, Ontario, Canada, 6 Department of Clinical Neurosciences, Department of Community Health Sciences, Cumming School of Medicine, University of Calgary, Calgary, Alberta, Canada, 7 Office of Public Health Studies, Thompson School of Social Work & Public Health, University of Hawaii at Mānoa Honolulu, Hawaii, United States of America, 8 Department of Physical Therapy, University of Alberta, Edmonton, Alberta, Canada, 9 Department of Research and Innovation, Institute of Disability and Rehabilitation Research, Ontario Tech University Toronto, Ontario, Canada, 10 Department of Health Research Methods, Evidence, and Impact, McMaster University (Hamilton, Ontario, Canada), 11 Health Sciences University, Bournemouth, United Kingdom, 12 Department of Rehabilitation Sciences, McMaster University, Hamilton, Ontario, Canada, 13 Department of Paediatrics, Pediatric Neurosciences, University of Alberta, Edmonton, Alberta, Canada, 14 Department of Pediatrics and Psychiatry, Faculty of Medicine & Dentistry, University of Alberta, Edmonton, Alberta, Canada

☯ These authors contributed equally to this work.
‡ JDC, MDH, ELH, GK, SM, QI, HT, MW, and JY also contributed equally to this work.
* svohra@ualberta.ca

**Data Availability Statement:** Data cannot be shared publicly because ethics approval did not include this nor did participant consent to this. Data are available from the Parker University Research

## Abstract

This prospective, community-based, active surveillance study aimed to report the incidence of moderate, severe, and serious adverse events (AEs) after chiropractic (n = 100) / physiotherapist (n = 50) visit in offices throughout North America between October-2015 and December-2017. Three content-validated questionnaires were used to collect AE information: two completed by the patient (pre-treatment [$T_0$] and 2–7 days post-treatment [$T_2$]) and one completed by the provider immediately post-treatment [$T_1$]. Any new or worsened symptom was considered an AE and further classified as mild, moderate, severe or serious. From the 42 participating providers (31 chiropractors; 11 physiotherapists), 3819 patient visits had complete $T_0$ and $T_1$ assessments. The patients were on average 50±18 years of age and 62.5% females. Neck/back pain was the most common presenting condition (70.0%) with 24.3% of patients reporting no condition/preventative care. From the patients visits with a complete $T_2$ assessment (n = 2136 patient visits, 55.9%), 21.3% reported an AE, of which: 7.9% were mild, 6.2% moderate, 3.7% severe, 1.5% serious, and 2.0% had missing severity rating. The most common symptoms reported with moderate or higher severity were discomfort/pain, stiffness, difficulty walking and headache. This study provides valuable information for patients and providers regarding incidence and severity of AEs following patient visits in multiple community-based professions. These findings can be used to inform

Repository (accessed via https://my.parker.edu/ICS/Research/Research_Data_Repository.jnz) for researchers who meet the criteria for access to confidential data.

**Funding:** The study was Canadian Institutes of Health Research, Alberta Innovates – Health Solutions, and the generosity of the Stollery Children's Hospital Foundation and supporters of the Lois Hole Hospital for Women through the Women and Children's Health Research Institute. The funders had no role in study design, data collection and analysis, decision to publish, or preparation of the manuscript.

**Competing interests:** The authors have declared that no competing interests exist.

patients of what AEs may occur and future research opportunities can focus on mitigating common AEs.

## Introduction

The United States's Institute of Medicine (now the National Academy of Medicine) has urged providers across health care settings to monitor adverse events (AEs) to improve patient safety [1]. The importance of monitoring AEs includes not only the identification of risks and preventable injury, but also providing patients with an accurate safety profile for treatments, which can contribute to setting more appropriate expectations and improving informed decision making about treatment options [1–3]. The logistics of identifying and reporting AEs in health care systems can be complex, for example, reported incidents usually have multiple potential causes; data collected rarely have sufficient information to assess causation; and there may be a poor understanding of underlying mechanism of injury/disease processes [4]. Like other complex settings (e.g., aerospace, nuclear power), health care systems require substantial resources to develop and maintain successful monitoring systems and a corresponding just and trusting, open, constructive environment where reporting harms is not punished [5].

However, the investigation of AEs in community-based health care settings is in its infancy. Reasons for this include: the lack of reporting systems that accurately collect AE information in these settings; heterogeneity in AE definitions; long periods between visits; and coordination of care among multiple providers [6]. In addition, the lack of a governance structure and administrative oversight in these settings compared to in hospitals, where institutional governance increases provider oversight and plays a major role in safe clinical care, is often convoluted and onerous [7, 8]. Regardless of setting, all regulated health care providers are overseen by their regulatory body; while patient safety is part of their mandate, regulatory colleges do not typically act unless a patient complains about the care received. For spinal manipulation therapy (SMT) providers in community-based settings, there is the added complexity of the natural history of presenting musculoskeletal (MSK) complaints, especially in regards to pain and stiffness that can be part of chronic, recurrent conditions that may wax and wane over time [9]. Given these known complexities, the first challenge in developing an AE monitoring system is to identify and report symptoms post-treatment [10]. Having an effective system to monitor post-treatment changes in symptoms can help to more accurately differentiate AEs from symptoms related to the conditions' natural history and/or can be used to predict symptom changes following different treatments, including those commonly used in ambulatory care settings by non-medical doctors, such as chiropractors and physiotherapists.

SMT is a popular treatment widely sought by many patients for MSK complaints, and most often delivered by chiropractors and physiotherapists [11, 12]. AEs associated with SMT have been studied using different research designs, including clinical trials [13, 14]. However, clinical trials are not the optimal design to collect rare AEs [15] and many observational studies lack standardized instruments and operational definitions for relevant terms [16]. AEs following SMT in adult patients have been most often reported as self-limiting, usually consisting of symptoms such as radiating MSK pain, nausea, dizziness, or tiredness [16–18]. More serious, but rare AEs such as cauda equina syndrome [18, 19] and stroke [20, 21] have also been investigated in well-designed case-control studies (i.e., large sample size, no recall bias or interviewer bias).

To help overcome the absence of high-quality prospective data on the frequency of AEs following SMT, the SafetyNet research program was developed. SafetyNet reflects the efforts of a

large international and multidisciplinary research team with expertise in chiropractic, physiotherapy, relevant medical specialties, research methodology, and patient safety [22]. One of the SafetyNet objectives aimed to develop and implement an active surveillance reporting system so that AEs could be reported to a central location and anonymously assessed. Details of the development of such a system has been reported elsewhere [23]. The primary aim of this study was to report the incidence of AE after patients being treated by chiropractors and physiotherapists, collected using the SafetyNet Active Surveillance Reporting System [22, 23].

## Materials and methods

This observational, prospective, community-based active surveillance study collected data between 19 October 2015 and 11 December 2017 in Canada and the United States of America (USA). To ensure similar data collection time periods, chiropractors were asked to collect data from 100 consecutive, unique patients, and physiotherapists, who self-reported commonly using SMT and were affiliated with the Canadian Academy of Manipulative Physiotherapy, were asked to collect data from 50 consecutive, unique patients since they provide SMT less commonly than chiropractors. Consecutive, unique patients were eligible whether they were new or returning patients, and were invited to participate only once during their provider's data collection period, regardless if SMT was provided at that visit. The University of Alberta Health Research Ethics Board reviewed and approved this study (Pro00021870). Each participating provider signed an informed consent prior to participating in this study. Participating patients received study information from participating providers (or their front desk) as well as from a study information letter outlining that informed consent was implied if they completed and returned the study questionnaires. To ensure confidentiality of patient's participation in this study, written and verbal consent was waived by the ethics committee.

### Patient and public involvement

Patients were involved in the design of the questionnaire described in the section named 'Data Collection'.

### Definitions

The operational definition for the study's primary outcome, reported AEs, was developed by an international, multidisciplinary team based on literature review and consensus with multiple stakeholders [23]. We defined AEs as any unfavorable sign, symptom or disease temporally associated with the treatment, whether or not caused by the treatment; specifically, any new or pre-existing symptom that is worse after treatment. Definitions of the four severity classifications self-rated by patients and practitioners and adjudicated centrally can be found in prior publications [23, 25, 26].

### Provider recruitment

Community-based chiropractors and physiotherapists in Canada and the USA were recruited through announcements at professional events and/or communications through their respective professional organizations, as well as social media, professional newsletters/magazines, and referrals from colleagues or past study participants. After enrollment, all providers completed a demographic questionnaire (including years of experience and number of patients per hour). They also had study materials (e.g., questionnaires) sent to them via mail; and, along with their clinic staff, received online one-on-one training on the study protocol by one of the study investigators (MF/KAP).

## Data collection

Information on AEs was collected by using three content-validated questionnaires, two completed by the patient and one completed by the provider (i.e., chiropractor or physiotherapist), with all questionnaires assessing symptoms the patient was experiencing at the time the questionnaire was completed. Content validity, considered the most important measurement property for patient reported outcome measures [24], was evaluated for all study instruments ($T_0$, $T_1$, provider long form, $T_2$) by a larger team of investigators and reported in detail elsewhere [23, 25, 26].

Symptoms assessed were: pain/discomfort, stiffness, weakness, fatigue/tiredness, headache, dizziness, numbness/tingling, nausea/vomiting, difficulty walking, problem sleeping, and "other". The patient completed a 'pre-treatment questionnaire' immediately prior to receiving their treatment (hereafter referred to as $T_0$). This questionnaire included questions about patients' demographic information, co-morbidities, use of medication and natural health products, the condition for which they were seeking care, and the symptoms they were experiencing at the time they were completing the questionnaire. After $T_0$, the provider delivered the treatment as they normally would to that patient. There were no modifications in care due to participation in the study.

Immediately after treatment (hereafter referred to as $T_1$), the provider completed the 'immediate post-treatment questionnaire'. $T_1$ information included what treatment(s) were provided, i.e., by type (manipulation, mobilization, mechanical device, other manual therapy, other non-manual therapy) and location (cervical spine, thoracic spine, lumbar spine, sacrum/pelvic, upper extremity, lower extremity, other). $T_1$ also included information regarding the symptoms currently identified by the patient or reported by the provider. These symptoms were rated by the provider as mild, moderate, severe, or serious using the study operational definitions. If rated as moderate, severe, or serious, the provider was asked to provide detailed information about the symptom and potential associated factors (hereafter referred to as provider long form). The information was sent and reviewed independently by two content experts (described below in *Adjudication for AE severity* section).

To maintain anonymity, only the patients were given the 'follow-up post-treatment questionnaire' to complete up to one week following treatment (hereafter referred to as $T_2$), together with a pre-addressed stamped return envelope to send the completed questionnaire directly to the study team. $T_2$ asked patients whether a reported symptom was new, better, worse, or unchanged since treatment, as well as satisfaction with care (1-very satisfied to 5-very unsatisfied) and if any additional care was sought to manage their symptoms in the time between the treatment and questionnaire completion. To establish symptom severity (mild, moderate, severe, or serious) at $T_0$ and $T_2$, patients were asked questions about symptom-related limitations based on the operational definitions.

## Adjudication for AE severity

For symptom severity established as moderate, severe, or serious at $T_1$, a provider long form was completed by the provider to give more detailed information about the event and potential associated factors. For the adjudication process, there were two blinded content experts: 1) an experienced (i.e., more than 10 years post-licensure) chiropractor (for AEs reported by chiropractors) or physiotherapist (for AEs reported by physiotherapists) and 2) an experienced physician with expertise in patient safety. Initially, one of the study investigators (MF/KAP) prepared a summary of the reported adverse events (AE) using information from the provider long-form and any additional relevant previously data from $T_0$, $T_1$ and $T_2$. The content experts then independently assessed the severity of the AE based on this summary. If the content

experts deemed the AE as moderate, severe, or serious, then the event was further evaluated by the content experts for causality/relatedness [23], preventability [27], and patient disposition [28].

## Data quality assurance and cleaning procedures

All data were entered and managed by a single data entry clerk using REDCap electronic data capture tools, hosted at the University of Alberta. REDCap is a secure, web-based application designed to support data capture for research studies [29, 30]. Data from the first 100 participants were verified by a single study investigator (MF) who compared $T_0$ and $T_1$ source documents to the data entered on REDCap to identify inconsistencies. Identified errors were communicated to data entry clerk. Subsequently, 20% of all entered data were randomly verified and validated by the same study investigator (MF). Data entry error rate was less than 0.5%. For audit purposes and to ensure transparency, all data entry changes were recorded with the time, date and user ID. Discussion of any query occurred with another study investigator (KAP) with the query resolutions recorded. After all data entry was completed, data were exported into an Excel spreadsheet, where duplicates were removed (n = 2), additional entry errors corrected (n = 5), and data cleaned.

## Sample size justification

The sample size estimation was based on two pilot SafetyNET studies (unpublished data). We observed 29 (5.33%) incidences of moderate to severe adverse events among 544 patient visits with intra-provider correlation of 0.1. No serious adverse event was reported. We anticipated an absolute precision of 2% on either side of the incidence. A sample size consisting of 53 providers each with 100 patient visits would produce a two-sided 95% confidence interval with a width of 4% when the incidence of adverse event is 5.33%. In the pilot studies, we observed about 50% non-participation and or dropout among initially interested providers and patients. To account for such loss, we aimed to enroll double the number of providers and their patients.

## Statistical analysis

We identified the incidence of AEs from three sources: 1) any new or worsening symptom reported by the provider on $T_1$; 2) any new or worsening symptom reported by the patient on $T_2$; or 3) any symptom on $T_2$ reported by the patient with increased severity in comparison to the same symptom on $T_0$ (irrespective of the patient self-reported symptom change, i.e., better, unchanged, worse, new). If a new symptom was reported on $T_2$, but the severity rating was missing, the symptom was considered worsened (i.e., an AE).

Patient demographics, presenting conditions, and health history were tabulated (mean with standard deviation or frequency with percentage). Incidence (%) of AEs either reported by patients, providers or both were calculated. Incidence of AEs were further classified according to symptoms and/or severity. For both classifications of AE and AE incidence, patients could report multiple AEs. Patients were considered to have an AE if they had one or more than one.

To assess type of data missingness, especially AE not missing at random, $T_0$ and $T_1$ were compared between patients who returned post-treatment form at $T_2$ vs. those who did not. A two-step procedure was used to impute missing data: 1) multiple imputation by chained equation (MICE) for missing covariates; [31–33] and 2) imputation of missing AEs using a mixed model. The covariates: age, sex, condition, treatment dose (i.e., number of treatments reported by the provider), treatment location, number of potential risk factors (e.g., patient's $T_0$ demographic and health history information), radicular pain, health insurance and providers' years

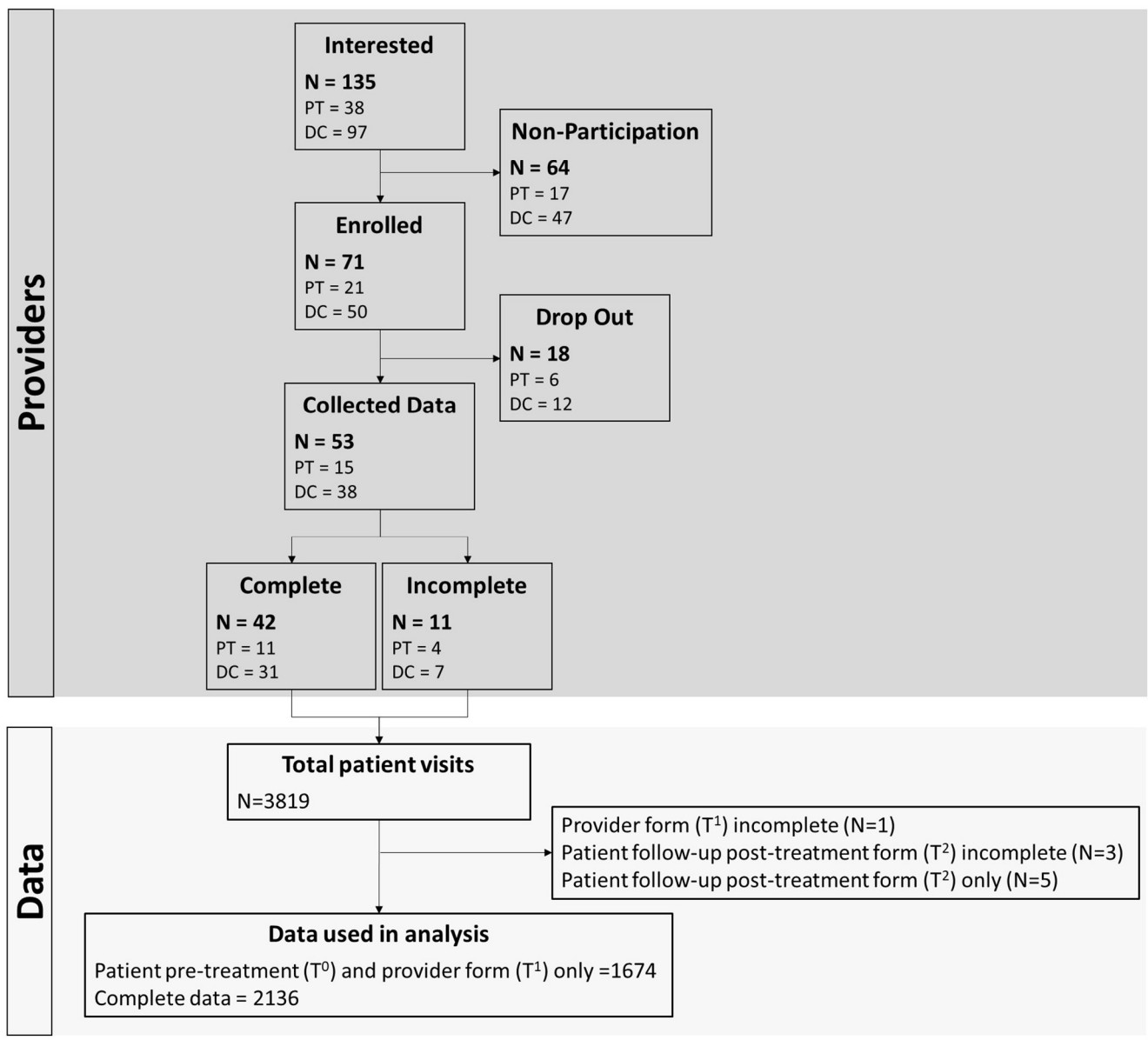

**Fig 1. STROBE diagram illustrating provider and data flow.**

of experience, and number of patients per hour were used in the imputation process. Logistic regression for binary variables (yes or no) and linear regression for continuous variables were used to impute missing covariate values. In each imputed data set, for participants with missing AE, probability of AE was predicted using a random intercepts (providers) logistic regression model. If the predicted probability of AE $> 0.5$, an incidence of AE was imputed. Otherwise, no AE was imputed.

## Results

Fig 1 shows participant and data flow. Specifically, 42 providers collected data from 3819 patient visits with complete data sets ($T_0$, $T_1$, and $T_2$) from 2136 patients (55.9%). The

**Table 1. Patient characteristics for participants who completed both study questionnaires (n = 2136).**

| Characteristics | Findings |
|---|---|
| Age in years (mean, ±SD) | 50 ± 18 |
| Sex (female; n, %) | 1336 (62.5) |
| Presenting condition^ (n, %)–more than one could be marked | |
| Neck/back pain | 1496 (70.0) |
| Extremity pain | 640 (30.0) |
| Other (e.g., headache) | 536 (25.1) |
| No condition–preventative | 518 (24.3) |
| Radicular pain (n, %) | 235 (11.0) |
| Chronicity (n, %) | |
| Acute (less than 3 weeks) | 356 (16.7) |
| Chronic (3 weeks or more) | 874 (40.9) |
| Health history (n, %) | |
| Musculoskeletal conditions (e.g., arthritis, prior spinal surgery) | 605 (28.3) |
| Metabolic conditions (e.g., diabetes, high cholesterol) | 418 (19.5) |
| Cardiovascular/Hematological conditions (e.g., bleeding disorder, stroke) | 42 (1.9) |
| Other (e.g., alcoholism, cancer, smoking, migraine, tuberculosis) | 748 (35.0) |
| Medication (n, %) | |
| None | 828 (38.8) |
| Aspirin | 238 (11.1) |
| Pain medication | 431 (20.2) |
| Other (e.g., Synthroid, blood pressure, Zoloft) | 800 (37.5) |
| Natural Health Products (n, %) | |
| None | 850 (39.8) |
| Garlic | 110 (5.1) |
| Omega-3 | 503 (23.5) |
| Vitamin E | 168 (7.9) |
| Other (e.g., turmeric, vitamin B, calcium) | 886 (41.5) |
| Payment (n, %) | |
| Self-pay | 1010 (47.3) |
| Car accident coverage | 93 (4.4) |
| Workers' Compensation | 21 (1.0) |
| Other insurance (e.g., private insurance/work coverage, partial) | 1066 (49.9) |

^- conditions may be episodic or wax/wane

providers (31 chiropractors; 11 physiotherapists) had an average of 18 years (SD: 10) of clinical experience and saw a median of 75 discrete patients per week (IQR: 125). Table 1 provides patient demographics. The average age of patients was 50 years (SD: 18), were more commonly female (62.5%), presented with MSK conditions (i.e., neck/back pain and/or extremity pain) (75.7%), and 24.3% reported the reason for their visit as "no condition/preventative care".

Tables 2 and 3 provide the incidence of AEs as reported by the patients and providers as new or worsening symptoms, and if self-reported, provider-reported, or identified from increased severity between $T_0$ and $T_2$. The overall incidence for all AEs was 21.3% per patient visit (n = 455/2136); the incidence of AE decreased to 6.3% (n = 12) in individuals with no prior symptoms, i.e., those who sought care for prevention/wellness with no presenting pain. Of all AEs reported, 7.9% (n = 168) were rated as mild in severity; 6.2% (n = 133) as moderate; 3.7% (n = 80) as severe; 1.5% (n = 33) as serious; and missing information did not allow for

**Table 2. Frequency and percentages of AEs (i.e. new or worsening symptoms) (n = 455 AEs reported) from provider, patient, and both (n = 2136 complete data sets).**

| | | n (%) |
|---|---|---|
| Patient Reported Only ($T_0$ & $T_2$) | Self-Assessed as Worse or New Symptom Only | 149 (7.0) |
| | Pre-Post Difference found Worsening Symptom Only | 104 (4.9) |
| | Symptom Reported by Both Self-Assessment & Pre-Post Difference | 35 (1.6) |
| Provider Reported Only ($T_1$) | | 127 (5.9) |
| Both Patient & Provider Reported ($T_0$, $T_1$, & $T_2$) | | 40 (1.9) |
| Total Visit with an AE Reported | | 455 (21.3)* |
| Total Visits without an AE Reported | | 1687 (78.7%) |

*Severity classification of total AEs were:

mild—168 (7.9%);

moderate—133 (6.2%);

severe—80 (3.7%); and

serious—33 (1.5%).

determining severity rating in 2.0% (n = 41). The majority of AEs were self-reported by the patient as worsening symptoms after the patient visit.

As shown in Fig 2, discomfort/pain and stiffness were the most common AEs (n = 152 and n = 87, respectively) with moderate, severe, and/or serious severity classification. These symptoms were usually reported as a worsening (i.e., pre-existing) symptom. The symptom reported least often was dizziness (n = 5); however, when it was reported, it was reported by the patient as worsening and as serious (i.e., led to hospitalization).

The $T_1$ form, which was completed by the provider immediately following the treatment visit, identified 127 AEs, of which 25 were rated by the provider as moderate, severe, or serious. Of these, only 13 (52.0%) had the provider long form completed as per protocol by 8 providers. Consensus from content experts was that all 13 AEs were mild; therefore, were not further assessed for causality/relatedness, preventability, or patient disposition. Providers failed to

**Table 3. The number of patients with an AE reported by provider and/or patient (n = 2136).**

| | Mild n (%) | Moderate n (%) | Severe n (%) | Serious n (%) | TOTAL Incidence/N (%) |
|---|---|---|---|---|---|
| Worsening AE—Provider-Reported | 26 (63.4) | 13 (31.7) | 2 (4.9) | 0 | 41/2136 (1.9) |
| Worsening AE–Increased severity rating on provider post | 0 | 74 (73.3) | 23 (22.8) | 4 (4.0) | 101/879 (11.5) |
| Worsening AE–Patient-Reported | 50 (41.3) | 39 (32.2) | 25 (20.7) | 4 (3.3) | 121/2136 (5.7) |
| Worsening AE–Increased severity rating on patient post | 0 | 67 (42.1) | 61 (38.4) | 31 (19.5) | 159/1280 (12.4) |
| New AE—Provider Reported | 36 (78.3) | 8 (17.4) | 2 (4.3) | 0 | 46/2136 (2.2) |
| New AE—Not marked on patient pre & not identified as better, worse, unchanged by provider | 8 (80.0) | 1 (10.0) | 0 | 0 | 10/2136 (0.5) |
| New AE–Patient-Reported | 55 (52.9) | 29 (27.9) | 12 (11.5) | 5 (4.8) | 104/2136 (4.9) |
| New AE—Not marked on patient pre & not identified as better, worse, unchanged by patient | 10 (33.3) | 7 (23.3) | 3 (10.0) | 2 (6.7) | 30/2136 (1.4) |

NOTE: Data based on comparison of patient PRE-form to provider POST-form and/or patient POST form.

Note: For rows 3, 6–8, the total AE was little higher than the stratification based on severity because among identified AE, severity was not known for 2% of them.

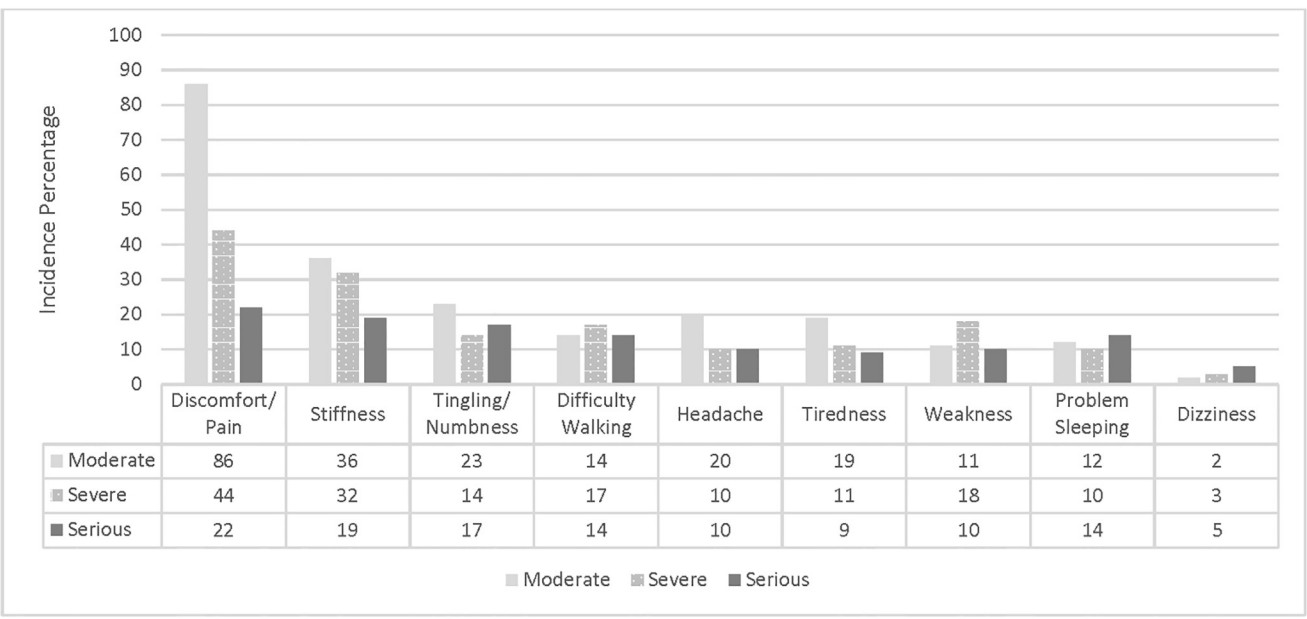

**Fig 2. Incidence (%) by severity (moderate, severe, and/or serious) for symptoms reported by patient and provider as either a worsening or new AE.**

complete the provider long form as per the study protocol for 12 AEs (48.0%) they reported as moderate or severe; these AEs could not be further assessed by the adjudication committee as insufficient information was available.

The frequency of AEs reported on $T_1$ were less among patients who did not return $T_2$ compared to the patients who did (8.7% vs. 11.5%). Patients whose symptoms were less severe at $T_0$ or improving at $T_1$ trended towards not returning the $T_2$ form. The observed evidence indicated AE data were not missing at random; as such, the two-step imputation procedure was conducted. Based on the multiple imputed data sets, the incidence of AEs was estimated to be 12.3% (95% CI: 11.2%–13.3%).

The assessment of possible bias due to patients not returning $T_2$, found that there were four variables, after adjusting for all variables, to be statistically associated with the patient returning of $T_2$: 1) for each additional potential risk factor (i.e., patient's $T_0$ demographic and health history information) a patient's odds of returning the $T_2$ increased by 16% (95% CI: 1.06, 1.27); 2) for each decreased number of patients that the provider saw per hour, the odds of the participating patient returning the $T_2$ increased by 5% (95% CI: 1.02, 1.10); 3) for each additional year the provider was in practice, the odds of the patient returning their $T_2$ increased by 3% (95% CI: 1.02, 1.04); and 4) the increased number of manipulation (high-velocity, low-amplitude or grade 5 mobilization) versus mobilization (grade 1–4) or other treatments given at the visit, the odds of the patient returning $T_2$ decreased by 2% (95% CI: 0.96, 1.00).

## Discussion

We found almost a fifth of patient visits to a chiropractor or physiotherapist reported an AE. These visits were primarily for MSK conditions (75.7% of presenting patients) and the majority of the AEs were of mild or moderate severity. The incidence of AEs reported in our study was lower than the 30%-50% reported in a recent scoping review of 250 observational and experimental studies of manual treatments of the spine [10]. Specifically, in 1996, a similar prospective clinic-based survey collected data from 4712 encounters from Norwegian

chiropractors found that 55% of these encounters had an AE [34]. In 2003, 280 patients in a clinical trial of chiropractic care for patients with neck pain found that 30% reported an AE [14]. In 2013, the Scandinavian College of Naprapathic Manual Medicine in Stockholm collected AE information from 767 patients and found that 51% of those who had at least 3 SMT treatments reported an AE [35].

There are many potential reasons for the differences in AE rates reported in studies. One is the heterogeneity in data collection including the timing of when AE information was sought. Prior to the start of our study, participating stakeholders recommended collecting AE information from the provider immediately after treatment but to not ask the patient until at least 48 hours later as it was felt that asking about AE immediately after treatment would identify common mild AEs that were expected after manual therapy and potentially deter providers from participating in the study. Other potential sources of heterogeneity include different levels and standards of training among the providers, the conditions for which patients sought care, cultural differences in reporting AE and litigation, differences in provider-patient communication/relationship, and use of nomenclature. Existing guidelines for creating optimal AE registries include the need to have standard nomenclature to code AEs to ensure accuracy and consistency, in addition to a systematic data collection system so that all AEs can be evaluated in a consistent and contemporaneous manner [36].

In regard to the severity classification, Swait & Finch's scoping review found that most of the AEs following manual treatments were transient and mild in nature [10]. Similarly, Senstad et al. described 85% of the AEs as mild or moderate [34]. In contrast, the severity mean range reported by Hurwitz et al. was between 1.0–6.0 (0–10 numerical scale) [14], while Paanalahti et al. reported the majority of the reports as 'short minor' [35]. Our study found that 7.9% were mild, with moderate or higher rated severity occurring with a frequency of 11.4%, which is lower than those reported by Senstad et al. 1997, but higher than those reported by Paanalahti et al. 2014. Heterogeneity in how severity classification information is collected and synthesized can impact comparability between studies. For transparency and comparability with future studies, we used case definitions to determine both new and worsening AEs, as well as a series of questions to determine severity.

It is important to consider AE profiles for other common treatments used for MSK conditions, such as aspirin, NSAIDS, and opioids. Daily aspirin use has a range of AE from mild (e.g., minor bruising) to severe (e.g., gastrointestinal bleeding) [37] and NSAIDS range of AE symptoms include cardiovascular issues, kidney injuries, and intestinal bleeding [38]. The harms associated with opioids have also been well described, including addiction and death [38].

The participating provider and patients in our study had similar demographic characteristics to those reported in a recent description of chiropractors' and physiotherapist practice profiles in Canada [39, 40]. The chiropractic practice profile study had an average of 15 years of experience and in our study, the mean years of experience was 18. While the physiotherapist profile did not directly report the years of experience, the proportion of practicing physiotherapist less than 50 years of age was 36.8%. The average patient in Mior et al., was between 45–64 years of age and female (59%) [39], Sutherland reported the largest age group was 45–59 years of age with no report on sex [40], and in our study the average age was 50.0 (SD: 18) and most commonly female (62.5%). The most common reason for seeking care or diagnoses were back pain and neck pain in both Mior et al. and our study. Thus, our findings may be generalizable to other chiropractors and physiotherapists practicing in similar jurisdictions.

Although considered rare, serious AEs, such as disc herniations, spinal cord injuries, dural tears with intracranial hypotension, phrenic nerve paralysis, or cauda equina, have been reported after SMT [10]. Implementation of established reporting systems for popular

interventions, such as SMT, would allow for rigorous prospective data collection, real-time investigation, and new learning opportunities. These reporting systems, also known as active surveillance systems, are often used with immunization programs and with drugs/devices in acute care. With enhanced electronic health record capacity, surveillance opportunities have increased, which can contribute to increasing intervention safety profile and public confidence, especially if used at a community-based level with multiple disciplines [41]. Identification of serious rare AEs could be enhanced through linkage between electronic health record data (e.g., linkage between community-based records and hospitalization data), which currently do not exist in Canada or the USA.

The SafetyNet Reporting System has been successfully used in chiropractic pediatric offices and a chiropractic teaching clinic [25, 26]. For the pediatric population seeking care within chiropractic offices for mostly MSK conditions, the overall incidence of AEs was found to be 8.8% (95% CI 6.72% to 11.18%) from 1179 unique pediatric (<14 years of age) patients [26]. Within the chiropractic teaching clinics, the AE incidence rate following patient visit was found to be 8.9% in a small sample of 89 patients seeking care for mostly MSK conditions [25]. While the AE incidence reported in these previous studies are slightly lower than that reported in our current study, both studies had smaller sample sizes and represented different clinical populations and settings. With the need for more community-based methods to enhance patient safety, including improving the identification and reporting of adverse events, future studies could model the SafetyNet Reporting System to assess incidence of AEs in other parts of the world, other treatment modalities, and other populations.

As one of the first and largest prospective evaluation of AEs following a chiropractic or physiotherapist visit, our study has an additional notable strength. A common limitation in patient safety literature is the lack of *a priori* definitions for AEs and severity classifications [42]. Throughout the literature, when these definitions are given, there is a vast range of descriptors with a lack of consistency both in terms of use and application, which impedes aggregating patient safety data for a more comprehensive picture [43, 44]. The definition used in our study was deliberately chosen to be conservative in nature, as determined by the multidisciplinary team of patient safety and SMT content experts to enhance confidence in study findings [23]. While more work is needed to develop standardized definitions for AEs among all SMT providers and is underway [45], this study demonstrates the successful implementation of a system to identify and collect AE data and classify according to agreed upon *a priori* definitions.

Another safety incident reporting system implemented specifically in the chiropractic profession, the chiropractic patient incident reporting and learning system (CPiRLS), was launched in 2009 in the United Kingdom [46]. A recent analysis of the incidents reported to CPiRLS since its launch found that this passive surveillance methodology had an extremely low number of incidents reported, which was similar to the cluster RCT that compared Safety-NET active surveillance to CPiRLS [25]. Passive surveillance may not be adequate to improve the identification and reporting of potential harms. However, CPiRLS identifies and disseminates key areas for patient safety improvement, which is an important part of patient safety [46]. CPiRLS also allows for indirect incidents to be reported, which are also vital to support a patient safety culture. Future endeavors to reporting incidents should continue to explore use of active and passive reporting system for both direct and indirect incidents.

Since our study is observational, we focused on association rather than causation. This distinction is particularly important for conditions with chronic and/or recurrent symptoms, such as musculoskeletal (MSK) disorders. Natural variation in MSK conditions (i.e., fluctuations in symptoms regardless of treatment) may be incorrectly attributed to treatment. Despite this limitation, systematic and meticulous documentation and thoughtful analysis of

observational data, is the foundation of advancing patient safety research. As a result, we feel the results from this study will help advance the field as it demonstrated successful data collection in multi-disciplinary community-based settings.

Our study methodology did not allow for comparison of patient and provider AE reports, as they were measured at different time points. Additionally, while patients were asked to complete the 2–7 days post-treatment [$T_2$] assessment before their next treatment visit, this was not monitored and not all patients provided adequate information to allow for AE severity to be determined, specifically,1683 participants due to missing $T_2$ form. As stated in the results, the forms were not missing at random, likely associated with less severe symptoms or improving patients. The missing data likely biased the estimation of AE upward as shown on the pragmatic two-step imputation approach. The imputed data suggested lower percentage of AE than observed data (12.3% vs 21.3%). Also, our sample size may not have been sufficient to capture serious rare AEs. Finally, the measurement properties of instruments developed for this study were evaluated for content validity only. While this has been found to be the most important measurement property [24], further assessment of how best to assess change in symptoms should be conducted to minimize recall bias and maximize accuracy.

## Conclusions

Our community-based active surveillance study found the incidence of AEs following chiropractic or physiotherapy patient encounters to be 21.3%. Of these AE reports, the severity classifications were noted as: mild (7.9%), moderate (6.2%), severe (3.7%), serious (1.5%), and missing severity responses (2.0%). This innovative and novel study provides valuable information for clinicians and patients and serves as a framework to more fully understand post-visits AEs more fully and potential strategies to mitigate them.

## Acknowledgments

We wish to thank all the participating providers and their patients. This study could not have been done without their support. We also acknowledge the numerous support personnel who ensured that providers were recruited, and data was collected accurately.

## Author Contributions

**Conceptualization:** Katherine A. Pohlman, Maeve O'Beirne, J. David Cassidy, Michael D. Hill, Eric L. Hurwitz, Gregory Kawchuk, Silvano Mior, Haymo Thiel, Michael Westaway, Jerome Yager, Sunita Vohra.

**Data curation:** Katherine A. Pohlman, Martha Funabashi, Maeve O'Beirne, Gregory Kawchuk, Sunita Vohra.

**Formal analysis:** Katherine A. Pohlman, Martha Funabashi, Maeve O'Beirne, J. David Cassidy, Eric L. Hurwitz, Silvano Mior, Quazi Ibrahim, Sunita Vohra.

**Funding acquisition:** Maeve O'Beirne, J. David Cassidy, Michael D. Hill, Eric L. Hurwitz, Gregory Kawchuk, Silvano Mior, Jerome Yager, Sunita Vohra.

**Investigation:** Katherine A. Pohlman, Martha Funabashi, Maeve O'Beirne, Sunita Vohra.

**Methodology:** Katherine A. Pohlman, Martha Funabashi, Maeve O'Beirne, J. David Cassidy, Michael D. Hill, Eric L. Hurwitz, Gregory Kawchuk, Silvano Mior, Quazi Ibrahim, Haymo Thiel, Michael Westaway, Jerome Yager, Sunita Vohra.

**Project administration:** Katherine A. Pohlman, Sunita Vohra.

**Resources:** Sunita Vohra.

**Software:** Sunita Vohra.

**Supervision:** Katherine A. Pohlman, Maeve O'Beirne, J. David Cassidy, Michael D. Hill, Eric L. Hurwitz, Gregory Kawchuk, Silvano Mior, Haymo Thiel, Michael Westaway, Jerome Yager, Sunita Vohra.

**Validation:** Katherine A. Pohlman, Martha Funabashi, Silvano Mior, Sunita Vohra.

**Visualization:** Sunita Vohra.

**Writing – original draft:** Katherine A. Pohlman, Martha Funabashi, Sunita Vohra.

**Writing – review & editing:** Katherine A. Pohlman, Martha Funabashi, Maeve O'Beirne, J. David Cassidy, Michael D. Hill, Eric L. Hurwitz, Gregory Kawchuk, Silvano Mior, Quazi Ibrahim, Haymo Thiel, Michael Westaway, Jerome Yager, Sunita Vohra.

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
