## [Decision Letter · Decision Letter 0]

21 May 2024

PONE-D-24-09373What’s the harm? Results of an active surveillance adverse event reporting system for spinal manipulation therapy providersPLOS ONE

Dear Dr. Pohlman,

Thank you for submitting your manuscript to PLOS ONE. After careful consideration, we feel that it has merit but does not fully meet PLOS ONE’s publication criteria as it currently stands. Therefore, we invite you to submit a revised version of the manuscript that addresses the points raised during the review process.

We look forward to receiving your revised manuscript.

Kind regards,

Holakoo Mohsenifar

Academic Editor

PLOS ONE

https://journals.plos.org/plosone/s/file?id=ba62/PLOSOne_formatting_sample_title_authors_affiliations.pdf"

2.Thank you for stating the following financial disclosure: 

"Funding for this study was received from the: 

Canadian Institutes of Health Research, 

Alberta References Innovates – Health

Solutions, and 

Stollery Children’s Hospital Foundation (supporters of the Lois Hole Hospital for Women through the Women and Children’s Health Research Institute)"

3. In this instance it seems there may be acceptable restrictions in place that prevent the public sharing of your minimal data. However, in line with our goal of ensuring long-term data availability to all interested researchers, PLOS’ Data Policy states that authors cannot be the sole named individuals responsible for ensuring data access (http://journals.plos.org/plosone/s/data-availability#loc-acceptable-data-sharing-methods).

Reviewers' comments:

Reviewer's Responses to Questions

**Comments to the Author**

1. Is the manuscript technically sound, and do the data support the conclusions?

Reviewer #1: Yes

Reviewer #2: Yes

Reviewer #3: Yes

2. Has the statistical analysis been performed appropriately and rigorously? 

Reviewer #1: Yes

Reviewer #2: Yes

Reviewer #3: Yes

3. Have the authors made all data underlying the findings in their manuscript fully available?

Reviewer #1: Yes

Reviewer #2: Yes

Reviewer #3: Yes

4. Is the manuscript presented in an intelligible fashion and written in standard English?

Reviewer #1: Yes

Reviewer #2: Yes

Reviewer #3: Yes

5. Review Comments to the Author

Reviewer #1: Some comments and suggestions for consideration by the authors:

Materials and Methods

Line 125: First sentence outlines time frame but not location. Would the authors consider adding country here? I appreciate it does appear later in the manuscript.

Adjudication for AE severity

Line194: commencing with “Firstly, a summary….” I found this sentence confusing and even after a number of re-reads, still found it difficult to comprehend. Would suggest the following for the authors’ consideration: Initially, one of the study investigators prepared a summary of the reported adverse events (AE) using information from the provider long-form and any relevant previously provided data. The content experts then assessed the severity of the AE independently based on this summary.

Statistical analysis

Line 235: post-treatment form at T2 vs “who did not” should read: those who did not.

Discussion

Line310: first sentence: consider the following: We found almost a fifth of patient visits to a chiropractor or physiotherapist reported an AE.

Reviewer #2: Thank you for the opportunity to review this manuscript which I think will be of interest to the readers of PlosOne. The authors are to be congratulated for their work and for contributing to this important area in manual therapy research. My comments and feedback are given with the intent to both clarify and strengthen this manuscript.

Title

Consider editing to remove the term spinal manipulation therapy (SMT) as there was no data reported to suggest that spinal manipulation (SM) was delivered to every participant. I argue that it is more appropriate to use the term manual therapy instead. If the authors are reluctant to do this, they should provide strong justification for the use of SMT in the title and clearly define in the manuscript what is meant by this and what treatment included i.e. not only SM but also (most likely) spinal mobilization and ancillary procedures (e.g. advice, stretching, trigger point therapy, dry needling etc). This data should be reported.

Abstract

Reads well but please check reported values for accuracy of reporting as some values in the main text Results section seem to be incorrect.

Introduction

- P3, line 88: would it be helpful to make this sentence more specific to manual therapy?

- P3, lines 89-91: 'severity and frequency' does not fit here or, needs further explanation

- P3, line 91: please clarify what is meant by 'oversight' i.e. by professional registration boards, governments, professional associations?

- P3, lines 88-94 & P4, lines 95-104: please consider acknowledging that there already exist other reporting systems (e.g. CPiRLS) in the literature and why the current reporting system is needed/relevant to your setting (and follow-up in the Discussion)

Methods and Materials

- P5, lines 126-129: please clarify if these numbers (i.e. 100/50) relate to unique patients or, patient visits and also if these numbers referred to all patients or, only patients who received SM. If it was all patients, did all receive SM? If not, this would support my argument to remove SMT from the title. If it was patient visits (not unique patients), did all participants return the questionnaire before their next visit?

- P5, line 128: please clarify if the clinicians self-reported using SMT or, how these data were obtained

- P6, line 156: who provided the one-on-one training? Was this person responsible for training all practitioners?

- P6, lines 170-171: similar to my comment re the title, please provide further details regarding what components 'treatment' could contain

- P6, lines 180-183: was there any trend regarding differences in reporting of AE at say 2 vs. 7 days? Could this have influenced the results if all participants returned their questionnaire at 7 days vs. 2 days? Similar to my comment above, for patients who had appointments close together (i.e. twice in one week), was the questionnaire always returned prior to their second visit?

- P7, lines 192-194: how was 'experienced' defined? Years of clinical experience?

- P7, lines 194-196: this sentence does not make sense to me, is it possible that a verb is missing?

- P8, line 238: how were 'treatment dose' and 'number of risk factors' determined/defined?

Results

- Table 1: what is the relevance of 'mark all that apply' in the presenting condition row? Why are neck and back pain reported together? Is there evidence of equal incidence rates following SM delivered to the neck and back? How were acute and chronic defined? Please provide examples for 'Musculoskeletal conditions', 'cardiovascular conditions', 'pain medication', 'medication – other', 'natural health products – other' and 'other insurance'.

- P10, line 264: this missing data should be discussed in the Discussion re influence it may have had on the results

- Table 2: is it possible to help orient readers by providing which questionnaires (i.e. T0/1/2) provided the data for each of the rows in this table?

- Table 3: please check the values reported in this table as by my calculation the total of four rows is not correct (rows 3, 6-8 of reported values)

- P11, line 280: please check dizziness n=5 as this is reported as n=10 in Figure 2 (if I understood the figure correctly)

- P11, line 301: similar to my previous comment, please provide an example of a 'risk factor'

Discussion

- P12, line 311: it is unclear where '75.7%' came from, is this from presenting condition in Table 1?

- P12, lines 326-327: '… levels and standards of training' of who? And what is meant by 'provider characteristics'?

- P14, lines 390-392: it is unclear what is meant by this sentence

- P14, lines 392-393: what is meant by 'natural variations in msk symptoms'?

- General comments:

- Please check references as I think P12, line 314 [11] is a typo and should be [10]

- Please consider discussing differences in incidence rates based on whether an active or passive reporting system is used

- Similar to my previous point, please consider discussing other existing AE reporting systems

- Parts of the Discussion read as quite general and I think could be improved with more specificity of the existing literature vs. the current study (e.g. what are frequencies of AEs for drug use or, if these are not available, please state that and why the paragraph discussion physiotherapists is relevant here when the inclusion criteria was that physios had to report frequent use of SM/SMT but I would argue that most physios do not apply SM unless they have specific training)

- In the paragraph reporting on the other SafetyNet studies, why do you not consider that the current results should be replicated in other adult cohorts? Do you see this system as having global generalizability or being more relevant to North America? Why/why not?

- How does the AE definition used in this study differ from existing definitions and do you think that might reduce comparability with the studies discussed in the Discussion section?

- Further discussion re sample size required as recruitment failed to reach calculated sample size

Reviewer #3: On line 63-64 and in Table 1 - you note the patients were 62.5% females. Capturing the patient characteristics of gender may be more inclusive in future work.

Line 268 reports severity classification of total AEs, however the numbers don't add up to 21.3 (2 missing). You do account for this in line 264 where you note that 2% had missing information. I think a line at 273 noting this missing information "missing severity information - 41 (2.0%)" would help the reader.

Table 2 (line 266) may benefit from an extra line "No AEs reported" to show the counterpart number to 21.3% (I presume there was no AEs in 78.7% of patient visits). A reader often looks to see that the numbers 'add up'.

Line 274, "Table 3. The number of patients with an AE reported by provider and/or patient (n=2136)" I thought 2136 was the number of patient visits that you had complete data sets for (T0+T1+T2). Should the number listed in line 274 not be (n=455)?

Line 278 "... were the most common AE (n=152 and n=87)". Should be AEs (you are discussing 2 categories)

Line 365 "especially if use at a community-based" should be 'used'

Line 369 "has been successfully used in pediatric offices and a chiropractic teaching clinic" - I think 'chiropractic' should be inserted before 'pediatric offices' for clarity.

6. PLOS authors have the option to publish the peer review history of their article (what does this mean?). If published, this will include your full peer review and any attached files.

Reviewer #1: **Yes: **Rosemary Giuriato

Reviewer #2: No

Reviewer #3: **Yes: **Christopher Burrell

---

## [Author Response · Author response to Decision Letter 0]

31 Jul 2024

1.Please ensure that your manuscript meets PLOS ONE's style requirements, including those for file naming. The PLOS ONE style templates can be found at:

 Authors’ Response: Documents have been named according to PLOS ONE’s style requirements. 

2.Thank you for stating the following financial disclosure: 

"Funding for this study was received from the: Canadian Institutes of Health Research, Alberta Innovates – Health Solutions, and Stollery Children’s Hospital Foundation (supporters of the Lois Hole Hospital for Women through the Women and Children’s Health Research Institute)"

 Authors’ Response: The cover letter has been changed as requested to reflect that the funders had no role in the study design, data collection and analysis, decision to publish, or preparation of the manuscript. 

3. In this instance it seems there may be acceptable restrictions in place that prevent the public sharing of your minimal data. However, in line with our goal of ensuring long-term data availability to all interested researchers, PLOS’ Data Policy states that authors cannot be the sole named individuals responsible for ensuring data access (http://journals.plos.org/plosone/s/data-availability#loc-acceptable-data-sharing-methods).

 Authors’ Response: As we did not anticipate this request, it was not part of our Health Research Ethics Board approval nor do we have participant consent for this. We have confirmed with our institution that we are unable to do this as requested. Data sharing can be considered via correspondence with Dr. Sunita Vohra (svohra@ualberta.ca).

Reviewers' comments:

Reviewer #1 - Some comments and suggestions for consideration by the authors:

Materials and Methods:

Line 125: First sentence outlines time frame but not location. Would the authors consider adding country here? I appreciate it does appear later in the manuscript.

 Authors’ Response: The location has been added (‘in Canada and the United States of America (USA)’) to line 125 as requested.

Adjudication for AE severity

Line194: commencing with “Firstly, a summary….” I found this sentence confusing and even after a number of re-reads, still found it difficult to comprehend. Would suggest the following for the authors’ consideration: Initially, one of the study investigators prepared a summary of the reported adverse events (AE) using information from the provider long-form and any relevant previously provided data. The content experts then assessed the severity of the AE independently based on this summary.

 Authors’ Response: Thank you – we have made the following change to this sentence:

‘Initially, one of the study investigators (MF/KAP) prepared a summary of the reported adverse events (AE) using information from the provider long-form and any additional relevant previously data from T0, T1 and T2. The content experts then independently assessed the severity of the AE based on this summary.’

Statistical analysis

Line 235: post-treatment form at T2 vs “who did not” should read: those who did not.

 Authors’ Response: This change has been made.

Discussion

Line310: first sentence: consider the following: We found almost a fifth of patient visits to a chiropractor or physiotherapist reported an AE.

 Authors’ Response: This change has been made.

Reviewer #2: Thank you for the opportunity to review this manuscript which I think will be of interest to the readers of PlosOne. The authors are to be congratulated for their work and for contributing to this important area in manual therapy research. My comments and feedback are given with the intent to both clarify and strengthen this manuscript.

Title

Consider editing to remove the term spinal manipulation therapy (SMT) as there was no data reported to suggest that spinal manipulation (SM) was delivered to every participant. I argue that it is more appropriate to use the term manual therapy instead. If the authors are reluctant to do this, they should provide strong justification for the use of SMT in the title and clearly define in the manuscript what is meant by this and what treatment included i.e. not only SM but also (most likely) spinal mobilization and ancillary procedures (e.g. advice, stretching, trigger point therapy, dry needling etc). This data should be reported.

 Authors’ Response: Thank you for this suggestion. To be as specific as possible, we changed ‘spinal manipulation therapy providers’ to ‘chiropractors and physiotherapists’. 

Abstract

Reads well but please check reported values for accuracy of reporting as some values in the main text Results section seem to be incorrect.

 Authors’ Response: Thank you. The values reported were all double-checked for accuracy with some percentages removed to align with reporting in the results section.

Introduction

- P3, line 88: would it be helpful to make this sentence more specific to manual therapy?

 Authors’ Response: The issues are larger than those pertaining only to manual therapy, hence our preference to draw attention to this by starting broadly (i.e., community-based settings) and then becoming more specific to our target population (line 96). 

- P3, lines 89-91: 'severity and frequency' does not fit here or, needs further explanation

 Authors’ Response: Removed ‘severity and frequency’

- P3, line 91: please clarify what is meant by 'oversight' i.e. by professional registration boards, governments, professional associations?

 Authors’ Response: This sentence has been modified to: 

‘In addition, the lack of governance structure and administrative oversight in these settings compared to in hospitals,...’

- P3, lines 88-94 & P4, lines 95-104: please consider acknowledging that there already exist other reporting systems (e.g. CPiRLS) in the literature and why the current reporting system is needed/relevant to your setting (and follow-up in the Discussion)

 Authors’ Response: Thank you. We have added the following paragraph about CPiRLS into the Discussion: 

‘Another safety incident reporting system implemented specifically in the chiropractic profession, the chiropractic patient incident reporting and learning system (CPiRLS), was launched in 2009 in the United Kingdom.[46] A recent analysis of the incidents reported to CPiRLS since its launch found that this passive surveillance methodology had an extremely low number of incidents reported, which was similar to the cluster RCT that compared SafetyNET active surveillance to CPiRLS.[25] Passive surveillance may not be adequate to improve the identification and reporting of potential harms. However, CPiRLS identifies and disseminates key areas for patient safety improvement, which is an important part of a patient safety culture.[46] CPiRLS also allows for indirect incidents to be reported, which are also vital to support patient safety. Future endeavors to reporting incidents should continue to explore use of active and passive reporting system for both direct and indirect incidents.’

Methods and Materials

- P5, lines 126-129: please clarify if these numbers (i.e. 100/50) relate to unique patients or, patient visits and also if these numbers referred to all patients or, only patients who received SM. 

Authors’ Response: The word ‘unique’ was added in several locations to try and clarify this issue: 

‘To ensure similar data collection time periods, chiropractors were asked to collect data from 100 consecutive, unique patients, and physiotherapists, who commonly reported using SMT, were asked to collect data from 50 consecutive, unique patients since they provide SMT less commonly than chiropractors. Consecutive, unique patients were eligible whether they were new, ongoing or returning patients, and were invited to participate only once during their provider’s data collection period, regardless if SMT was provided at that visit.’

If it was all patients, did all receive SM? If not, this would support my argument to remove SMT from the title. If it was patient visits (not unique patients), did all participants return the questionnaire before their next visit?

 Authors’ Response: 79.7% of participants received SMT. As suggested, we have modified the title to clarify that the study focused on SMT providers, not that SMT was received by all patients. Our study was unique patients, which has been clarified as per the comment above. 

- P5, line 128: please clarify if the clinicians self-reported using SMT or, how these data were obtained

 Authors’ Response: Modified to: ‘…who self-reported commonly using SMT,…’

- P6, line 156: who provided the one-on-one training? Was this person responsible for training all practitioners?

 Authors’ Response: Added the additional information: ‘by one of the study investigators (MF/KAP).’

- P6, lines 170-171: similar to my comment re the title, please provide further details regarding what components 'treatment' could contain

 Authors’ Response: More information was added here: 

‘T1 information included what treatment(s) were provided, i.e., type (manipulation, mobilization, mechanical device, other manual therapy, other non-manual therapy) and location (cervical spine, thoracic spine, lumbar spine, sacrum/pelvic, upper extremity, lower extremity, other).’

- P6, lines 180-183: was there any trend regarding differences in reporting of AE at say 2 vs. 7 days? Could this have influenced the results if all participants returned their questionnaire at 7 days vs. 2 days? 

 Authors’ Response: There was no difference based on when the AE was reported (data not shown). 

Similar to my comment above, for patients who had appointments close together (i.e. twice in one week), was the questionnaire always returned prior to their second visit?

Authors’ Response: Unfortunately, there was no way to monitor this. This was added into the limitations: 

‘…while patients were asked to complete the 2-7 days post-treatment [T2] assessment before their next treatment visit, this was not monitored…’

- P7, lines 192-194: how was 'experienced' defined? Years of clinical experience?

 Authors’ Response: This has been added for clarity: 

‘…an experienced (i.e., more than 10 years post-licensure) chiropractor…’

- P7, lines 194-196: this sentence does not make sense to me, is it possible that a verb is missing?

 Authors’ Response: This sentence has been modified for clarity.

- P8, line 238: how were 'treatment dose' and 'number of risk factors' determined/defined?

 Authors’ Response: Added additional information for clarity: 

‘…treatment dose (i.e., number of treatments reported by the provider),…’ 

‘number of potential risk factors (i.e., patient’s T0 demographic and health history information)’

Results

- Table 1: what is the relevance of 'mark all that apply' in the presenting condition row? 

 Authors’ Response: To clarify that more than one presenting condition could be identified, the following was added for clarification: ‘more than one could be marked’ 

Why are neck and back pain reported together? 

 Authors’ Response: For simplicity, we prefer to report data related to the spine vs. extremities. Future work will consider more detailed analysis. 

Is there evidence of equal incidence rates following SM delivered to the neck and back? 

 Authors’ Response: As we developed an a priori analysis plan for the primary outcome, we prefer to report according to it. We agree this is an interesting secondary question and will plan to look at it in future post hoc analyses. 

How were acute and chronic defined? 

 Authors’ Response: Information on definition has been added: 

 Acute (less than 3 weeks) 

 Chronic (3 weeks or more)

Please provide examples for 'Musculoskeletal conditions', 'cardiovascular conditions', 'pain medication', 'medication – other', 'natural health products – other' and 'other insurance'.

 Authors’ Response: All of these items have been updated with examples as provided on the study forms: 

‘Musculoskeletal conditions (e.g., arthritis, prior spinal surgery)’ 

‘Cardiovascular/Hematological conditions (e.g., bleeding disorder, stroke)’

‘Medication, Other (e.g., Synthroid, blood pressure, Zoloft)’

‘Natural Health Products, Other (e.g., turmeric, vitamin B, calcium)

‘Payment, Other insurance (e.g., private insurance/work coverage, partial)’

NOTE: ‘pain medication’ was whatever the patient was taking that they determined was for pain. 

- P10, line 264: this missing data should be discussed in the Discussion re influence it may have had on the results. 

 Authors’ Response: Thank you for your suggestion. The following few sentences were added in the discussion section:

‘Additionally, while patients were asked to complete the 2-7 days post-treatment [T2] assessment before their next treatment visit, this was not monitored and not all patients provided adequate information to allow for AE severity to be determined, specifically, 1683 participants due to missing T2 form. As stated in the results, the forms were not missing at random, likely associated with less severe symptoms or improving patients. The missing data likely biased the estimation of AE upward as shown on the pragmatic two-step imputation approach. The imputed data suggested lower percentage of finding less AE than observed data (12.3% vs 21.3%).’

- Table 2: is it possible to help orient readers by providing which questionnaires (i.e. T0/1/2) provided the data for each of the rows in this table?

 Authors’ Response: Thank you for this suggestion, which has been added to the table:

 n (%)

Patient Reported Only (T0 & T2) Self-Assessed as Worse or New Symptom Only 149 (7.0)

 Pre-Post Difference found Worsening Symptom Only 104 (4.9)

 Symptom Reported by Both Self-Assessment & Pre-Post Difference 35 (1.6)

Provider Reported Only (T1) 127 (5.9)

Both Patient & Provider Reported (T0, T1, & T2) 40 (1.9)

Total 455 (21.3)*

- Table 3: please check the values reported in this table as by my calculation the total of four rows is not correct (rows 3, 6-8 of reported values)

 Authors’ Response: Thank you for noting. For rows 3, 6-8, the total AE was little higher than the stratification based on severity because among identified AE, severity was not known for 2% of them. We mentioned the missing severity data in the results section and in the abstract. We added a footnote for Table 3 for the clarification. 

- P11, line 280: please check dizziness n=5 as this is reported as n=10 in Figure 2 (if I understood the figure correctly)

 Authors’ Response: This number reflects the combination of moderate, severe, and/or serious severity 

---

## [Editor Report · Decision Letter 1]

6 Aug 2024

What’s the harm? Results of an active surveillance adverse event reporting system for chiropractors and physiotherapists

PONE-D-24-09373R1

Dear Dr. Katherine A. Pohlman,

We’re pleased to inform you that your manuscript has been judged scientifically suitable for publication and will be formally accepted for publication once it meets all outstanding technical requirements.

Kind regards,

Holakoo Mohsenifar

Academic Editor

PLOS ONE
---

## [Editor Report · Acceptance letter]

8 Aug 2024

PONE-D-24-09373R1 

PLOS ONE

Dear Dr. Pohlman, 

I'm pleased to inform you that your manuscript has been deemed suitable for publication in PLOS ONE. Congratulations! Your manuscript is now being handed over to our production team.

Kind regards, 

on behalf of

Dr. Holakoo Mohsenifar 

Academic Editor

PLOS ONE